# Emulation of high-resolution land surface models using sparse Gaussian processes with application to JULES

Evan Baker[1], Anna Harper[2], Daniel Williamson[2], and Peter Challenor[2]

[1]Living Systems Institute, University of Exeter, Exeter EX4 4QD, UK
[2]Department of Mathematical Sciences, University of Exeter, Exeter EX4 4QF, UK

**Correspondence:** Evan Baker (e.baker@exeter.ac.uk)

**Abstract.**

Land surface models are typically integrated into global climate projections, but as their spatial resolution increases the prospect of using them to aid in local policy decisions becomes more appealing. If these complex models are to be used to make local decisions, then a full quantification of uncertainty is necessary, but the computational cost of running just one full
simulation at high resolution can hinder proper analysis.

Statistical emulation is an increasingly common technique for developing fast approximate models in a way that maintains accuracy but also provides comprehensive uncertainty bounds for the approximation. In this work, we develop a statistical emulation framework for land surface models, enabling fast predictions at a high resolution. To do so, our emulation framework acknowledges, and makes use of, the multitude of contextual data that is often fed into land surface models (sometimes called
forcing data, or driving data), such as air temperature or various soil properties. We use The Joint UK Land Environment Simulator (JULES) as a case study for this methodology, and perform initial sensitivity analysis and parameter tuning to showcase its capabilities. JULES is perhaps one of the most complex land surface models, and so our success here suggests incredible gains can be made for all types of land surface model.

## 1 Introduction

Land surface models (LSMs) represent the terrestrial biosphere within weather and climate models, focusing on hydrometeorology and biogeophysical coupling with the atmosphere. The latter includes nutrient flows between vegetation and soils, and the turbulent exchange of CO2, heat, moisture, and momentum between the land surface and the atmosphere. The Joint UK Land Environment Simulator (Cox et al., 1998; Best et al., 2011; Clark et al., 2011) is an example of an LSM; used for a variety of applications and temporal/spatial scales as part of the UK Met Office's Unified Modelling system. LSMs can be used
to further scientific understanding of land surface processes and to inform policy decisions. For both applications, increased confidence in simulated results and knowledge of model uncertainty is needed, which typically involves running the model many times with varied forcings and parameters (Booth et al., 2012; Murphy et al., 2004). The computational cost of running these models limits the number of runs that can be obtained, constraining the resulting analysis.

An important factor in computational cost versus practical relevance is the resolution at which the model can be run. Whilst there is a general, and not always justifiable, push towards higher resolution across climate modelling, when using LSMs to support landscape decisions (whether these be local/national government policy decisions, or landowner investment decisions), ensuring that the model is able to inform at decision relevant resolutions is critical. For example, if considering different policies to incentivise farmers to alter land use (by giving over land to tree planting for example), LSM simulations run to help understand the efficacy and risks of different policies would need to be at a high enough resolution to capture areas the size of individual farms (say at 1km). This can be incredibly costly; in Ritchie et al. (2019), at 1.5km × 1.5km resolution for Great Britain (77980 grid cells), JULES took approximately 25.5 hours to simulate a decade (with 20 years of spin up simulation to allow any input parameters to influence the present-day land surface, which takes approximately 19 additional hours) on 72 processors of the UK NERC/Met Office MONSooN supercomputer. Quantifying uncertainty for even a single policy option, let alone a diverse array of policies, would not be feasible using high resolution JULES directly.

Statistical surrogate models, also known as emulators, have been developed to combat this issue (Sacks et al., 1989; Kennedy and O'Hagan, 2001). An emulator is a statistical model that, once built, facilitates fast predictions of the output of a computer model, with quantified uncertainty in the predictions, and without any further simulation. The resulting statistical model provides a powerful tool for exploring, understanding, and improving the process-based model from which it is built. Fast predictions of land surface models could enable better decision-making, improve scientific understanding, and enable the effective linking of multiple models, all while quantifying the various uncertainties involved.

Emulators have been widely used by the climate community. They have been used to study the Met Office's coupled models (Williamson et al., 2013), including developing the UK Climate Projections in 2009 and 2018 (Sexton et al., 2012, 2019). Hemmings et al. (2015) build mechanistic emulators for specific locations for an ocean biogeochemical model; Petropoulos et al. (2014) conduct a comprehensive sensitivity analysis using an emulator for a land surface model; and Williamson et al. (2017) calibrates an ocean model using emulators. Such efforts typically only emulate a few key locations or summary statistics of interest, for example, McNeall et al. (2020) emulate (and calibrate) JULES considering only three averaged locations. These types of analysis can be useful in understanding the sensitivity of the model output to its different parameters, and for constraining parameter space, but cannot be used as surrogates to the full model when needed to support local decision-making.

Emulating spatio-temporal output in order to use an emulator as a surrogate to the full model has also received attention. Lu and Ricciuto (2019) attempt to emulate an LSM at a higher resolution, where they reduce the dimensionality of the output via singular value decomposition. This is a well-known strategy (Higdon et al., 2008), but lowering the dimension in this way can lead to a loss of information and interpretability, with documented negative effects (Salter et al., 2019). Additionally, Lu and Ricciuto (2019) use a neural network to construct their emulator. Whilst neural networks can be capable tools, they do not provide a complete quantification of the various uncertainties, which can be an essential quality when dealing with complex LSMs.

Attempts to emulate spatio-temporal models without dimension reduction, but by emulating each point in space and time separately, are also popular. Gu and Berger (2016) assume the same correlation function and parameters are shared across all outputs, enabling huge numbers of emulators to be built in parallel. The "emulate every grid cell" approach has been used

in environmental applications including aerosol models (Lee et al., 2012; Johnson et al., 2020), tsunami models (Salmanidou et al., 2021) and volcano models (Spiller et al., 2020). These methods do not directly model spatial and temporal dependence, instead assuming the training data for each grid cell contains enough information alone, and relying on the fast training and prediction times of standard emulators (though for JULES, the millions of emulators required to follow this approach renders it less feasible). Another approach is to consider space and time as inputs to the model, though this can render the number of data points too large for traditional Gaussian process implementations. Rougier (2008) uses separability in space and time to develop efficient emulators of this type that rely on kronecker identities for their speed.

Land surface models typically do not exchange information laterally between grid cells (LSMs with more sophisticated hydrology schemes can be an exception to this, where groundwater and rivers can flow between grid cells). Therefore, the structure of the spatio-temporal outputs is often controlled entirely by a set of pre-known forcing data (which could be observational data, or outputs from an atmospheric model, or pre-selected by a practitioner). For example, JULES relies on a set of driving data; providing information about the weather on a sub-daily time step, and various soil properties. In many cases, the land surface model output can be treated as independent in space and time, *conditional on the forcing data*. With this framing, the land surface model only outputs a spatial-temporal map because it is input a spatial-temporal map. In other words, grid cells in LSMs often do not "talk" to each other: what happens in one grid cell has no bearing on what happens in a neighbouring grid cell (except that their forcing data is likely to be similar, and that is why their outputs are likely to be similar).

In this paper we outline a framework for building emulators of LSMs, leveraging this interesting property of many LSMs to facilitate the emulation of the high-dimensional output. This framework is described in Section 2. We demonstrate the capabilities of such an emulator by emulating JULES at 1km resolution in Section 3. We also use this emulator to obtain a ranking of the relative importance of different model inputs, which can be key to scientific understanding and can guide future model development, and we use the emulator to tune the model via history matching. We offer discussion in Section 4 about the importance of this work and where improvements could be made; along with other possible applications of LSM emulators and future research opportunities.

## 2    Methods

### 2.1    Emulators

Gaussian processes (GPs) are commonly used to build emulators. Because complex models (simulators) can be slow to evaluate, they can be treated as a function that we are uncertain about. A Gaussian process provides a probability distribution over all possible functions that could recreate an observed simulation ensemble. These possible functions can then be quickly sampled, or evaluated, to predict the simulator output for new input values, while quantifying the respective uncertainty. Off-the-shelf software for implementing Gaussian processes is common place, including (but by no means limited to) the `DiceKriging` and `RobustGaSP` packages in **R** (Roustant et al., 2012; Gu et al., 2022), the `scikit-learn` and `GPy` packages in python (Pedregosa et al., 2011; GPy, 2012), and via the built-in Statistics and Machine Learning Toolbox in MATLAB. A Gaussian

process emulator can be written as:

$$y(\theta) \sim GP(m(\theta), k(\theta, \theta)) \tag{1}$$

where $y$ is the simulation output, $\theta$ is a vector of input parameters, m is a mean function that can be used to provide prior beliefs about the model, and $k$ is a covariance function.

Formally, a GP is a stochastic process such that any finite collection of outputs, $y(\theta_1), \ldots, y(\theta_n)$, has a multivariate normal distribution with mean $(m(\theta_1), \ldots, m(\theta_n))$ and variance matrix $K$, where $K_{ij} = k(\theta_i, \theta_j)$. After observing an ensemble of runs, **y**, the posterior process, $y(\theta) \mid$ **y**, is still a GP, and the mean and covariance function of this posterior GP have a known analytical form (Rasmussen and Williams, 2006). The prior mean function $m$ and covariance function $k$ have to be specified in advance. In this paper, we use a zero function $m$ (providing no prior beliefs about the shape of the output), and a standard non-isotropic squared exponential covariance function (sometimes known as an "automatic relevance determination kernel").

Throughout this article, emulating output $y$ and inputs $\theta$ with a Gaussian process is shortened to: $y(\theta) \sim GP(\theta)$. It can be good practice to re-scale the inputs to all be in [0,1], and to standardise the outputs (subtract the ensemble mean and divide by the ensemble standard deviation), and this is also done in this work.

## 2.2 Emulating a Land Surface Model

We consider the case of an LSM which outputs spatial maps that vary in time. Spatio-temporal maps can be very high dimensional; for example, our study area is Great Britain, where there are 230,615 1 km $\times$ 1 km grid cells. If Gross Primary Productivity (GPP) was output daily, then the simulation result for a single year would be an approximately 8.5 million dimensional output.

The spatio-temporal correlation structure in an LSM is often inherited through its input forcing data. For example, JULES solves the same differential equations independently in each grid cell and no information is passed horizontally by the solver. As such, we choose to treat an $S \times T$ dimensional output as $ST$ different 1D outputs, each with a different set of forcing data inputs (where $S$ represents the number of spatial locations and $T$ the number of timesteps). Mathematically, that is:

$$y_{ts}(W, \theta) = f(W_{ts}, \theta), \tag{2}$$

where $y_{ts}(W, \theta)$ is the output at time $t$ and location $s$, with forcing data $W$ and input parameters $\theta$, $f$ is the land surface model, and $W_{ts}$ is the forcing data only at time $t$ and at location $s$. By treating the LSM as a model with 1D output, a standard emulator can be built; that is:

$$f(W_{ts}, \theta) \sim GP(W_{ts}, \theta) \tag{3}$$

where $W_{ts}$ is treated like additional input parameters, along with $\theta$.

Whilst the assumption of spatial independence (conditional on the forcing data) can be sensible, temporal independence is often less so. In JULES, various internal state variables are stored and updated at each time step, which provides some temporal structure. For example, the soil moisture (which is modelled by JULES) depends not only on the precipitation at the current

time step, but also previous time steps. Another example is the leaf area index: future carbon assimilation and respiration depend on the leaf area index from the previous time step.

Such time structures could be emulated using a 'dynamic emulator' (Mohammadi et al., 2019), but the computational cost of this is prohibitive for the lengths of time LSMs often deal with. In this work, for JULES, we mitigate this time-dependence by working with 8-day averages, which mitigates the short term temporal dependence (for both the output and the forcing data). To control for some longer-term temporal dependencies, we supplement the forcing data $W_{ts}$ with a pseudo-forcing variable that records the day of the year. In this paper, this supplementary "day of the year" forcing input arbitrarily takes the $5^{th}$ day in the 8-day average (so the value of "day of the year" for data from January $1^{st}$ to January $8^{th}$ would be 5). This is then included as an additional column of $W_{ts}$. This acts as a proxy for any long-term (i.e. seasonal) temporal structure, which our assumption of time-independence ignores.

Any residual temporal correlation is then modelled as residual noise; and so the emulator is no longer forced to perfectly interpolate each simulation. This additional noise is sometimes called a 'nugget', and including it has been found to provide improved emulator performance regardless of whether it is technically needed (Gramacy and Lee, 2012).

An overview of this process is provided in Figure 1, outlining the various steps needed to build such a land surface emulator.

## 2.3 Data abundance

The formulation discussed previously, where each grid cell and each time step is treated as an independent (conditional on the forcing data) data point, results in tens of millions of data points per year for even a single simulation. Theoretically, an abundance of data should greatly improve predictive capabilities. However, Gaussian processes are not designed for large data sets, as they require the inversion of an $n \times n$ covariance matrix, where $n$ is the number of data points, and so computational time scales with $n^3$. In computer experiments, roughly 10 data points per input dimension is normally expected and recommended (Loeppky et al., 2009), and so tens of millions of data points is far beyond standard. We could use other statistical models here instead (such as a standard linear regression model, or a neural network), but the flexibility and the uncertainty estimates of a GP are desirable.

The concept of sparse Gaussian processes was developed to mitigate the computational issues of GPs (Snelson and Ghahramani, 2006). In effect, the idea is to learn some hypothetical, smaller, ideal data set, and use that to build the emulator. If $X$ is a set of inputs (the combined set of $W_{ts}$ and $\theta$) and $\mathbf{y}$ is a set of outputs, both $n$ in number, then the goal is to learn some set of inputs $Z$ and some set of outputs $\mathbf{u}$ which are only $M$ in number (where $M << n$) whilst maintaining as much accuracy as possible. The smaller set of data points $Z$ and $\mathbf{u}$ are called "inducing points". The input values of these inducing points do not need to be constrained as a subset of $X$, and so they can be freely placed at key locations in the input space of $X$. Learning these inducing points, along with all other parameters in the GP, can be done efficiently via variational inference (Titsias, 2009; Hensman et al., 2013, 2015), which involves optimising a lower bound for the likelihood of the data. A specific lower bound used for GPs also facilitates additional speed up by allowing small, random, batches of the total dataset (minibatches) to be used at each optimisation step. We make use of the python package GPflow (version 1.5.1) (Matthews et al., 2017) which provides

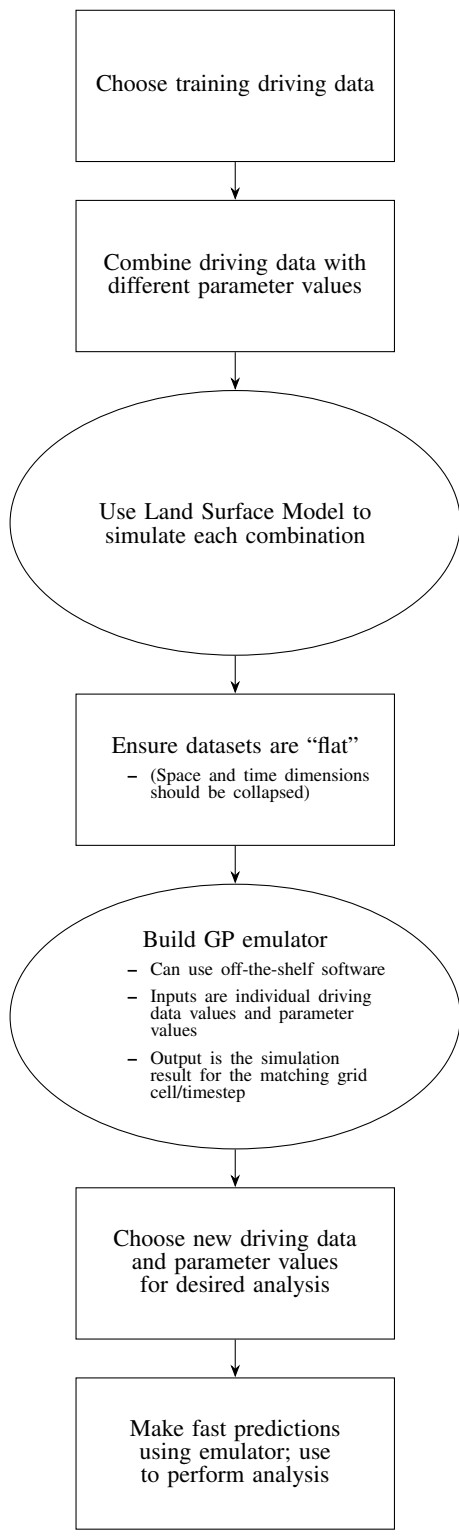

**Figure 1.** A flowchart broadly showing the steps needed to build and use a land-surface emulator under our framework.

an implementation for sparse Gaussian processes. For the GPflow settings, we used 10 inducing points per input dimension and a minibatch size of 1000. Further details for sparse GPs are provided in Appendix A, and in the given references.

## 2.4    Obtaining an Ensemble

To build an emulator of the type described previously, we need several simulations from the land surface model. Standard practice would be to run the land surface model for the entire area of interest (for example, Great Britain), and then repeat this

several times with different parameter values to obtain an initial ensemble (Murphy et al., 2004; Booth et al., 2012; Williamson et al., 2017). Our assumption of LSM grid cell independence means that we do not need to run the LSM for the entire area of interest. Instead, we can select only a subset of coordinates, and pair each with a different parameter setting. For JULES, we select the subset of grid cell coordinates so that they, collectively, are representative of the different regimes of forcing data that appear in Great Britain. We then paired each coordinate with its own parameter setting in such a way that the set of $(W_{ts}, \theta)$

combinations were sufficiently diverse. Details for both selection processes are provided in Appendix B.

     To see the potential benefits of this approach, consider an example where the area of interest contains 200,000 grid cells, and we could afford to run the LSM for this area 10 times, each time with a different $\theta$ value. Alternatively, we could run 2,000,000 different $\theta$ values, each one run at only one coordinate. Collectively, this second option should still provide a good coverage of the different possible forcing data, but we would now have *much* better coverage in $\theta$, without requiring any additional

simulation effort.

## 2.5    Emulating JULES

To demonstrate the outlined framework, we build an emulator for JULES. We narrow our focus to only investigate Gross Primary Productivity (GPP), which is a measure of plant photosynthesis. We begin with GPP as this is the entry point of carbon into the terrestrial carbon cycle. Further work could focus on emulating other aspects of the land carbon cycle (such

as Net Primary Productivity, or vegetation and soil stores). The carbon cycle is relevant for studies of climate change, and for these applications JULES is typically run globally with a course spatial resolution (at least 0.5° x 0.5°). However our study is motivated by an increasing need for detailed process models to inform decisions about land use and management at a much higher resolution. Therefore, we run and emulate JULES at a 1km × 1km resolution for Great Britain[1].

     Grid cells in JULES are subdivided into tiles representing vegetated and non-vegetated surfaces. On vegetated tiles, JULES

calculates GPP for different plant functional types (PFTs). The grid cell GPP is a weighted average of the PFT-dependent GPP (depending on the fractional area covered by each PFT). The 5 PFTs we use are: deciduous broadleaf trees (BT), evergreen needleleaf trees (NT), C3 grasses (C3g), shrubs (SH) and cropland (Cr). The fractional area covered by each PFT is set based on land cover data (see Appendix C for details on the JULES simulations). We build 5 independent emulators, one for each

---

[1]At a 1 km × 1 km resolution, our setup for Great Britain has a total of 230,615 grid cells.

PFT. These independent emulators can then be summed together, weighted by the PFT fractions, to provide the final emulator for overall GPP. In summary, the overall emulator is then:

$$\text{GPP}(W_{ts}, \theta) = \sum_{j \in \{PFT\}} \gamma_j \text{GPP}_j(W_{ts}, \theta_j); \tag{4}$$

with

$$\text{GPP}_j(W_{ts}, \theta_j) \sim GP(W_{ts}, \theta_j), \tag{5}$$

where $\gamma_j$ is the fraction of PFT $j$ in the grid cell, $\text{GPP}_j$ is the PFT-$j$-specific GPP value, and $\theta_j$ is the collection of input parameters that are relevant for PFT $j$. This overall emulator is simply the sum of 5 distinct emulators, one per PFT. Building one emulator for each PFT and then summing them together makes use of more information known to JULES, which should improve the accuracy of the overall emulator. It also reduces the dimension of $\theta$ in each individual emulator.

The PFTs are characterised by parameters describing their physiology, radiative properties, seasonal responses and other attributes. In this configuration of JULES, there are approximately 50 relevant parameters for each PFT. We investigate only 13, 10 of which are different for each PFT, chosen based on previous sensitivity studies of JULES (Booth et al., 2012; Raoult et al., 2016), and experience working with JULES. Table 1 lists the different tuning parameters $\theta$, standard values, the ranges we consider, and their role in JULES.

Table 1: A table showing the parameters, their "standard" values, their initial un-tuned ranges, and a short description. Most parameters are given different ranges for each plant functional type, indicated by the brackets in the parameter name, these are broadleaf trees (BT), needle-leaf trees (NT), C3 grasses (C3g), shrubs (SH), and crops (Cr).

| Parameter | Standard | Range | Description |
|---|---|---|---|
| alpha | 0.08 | $[0.04, 0.12]$ | Quantum efficiency of photosynthesis (mol $CO_2$ (mol PAR photons)$^{-1}$) |
| knl | 0.2 | $[0.05, 0.35]$ | Rate of decay of $N$ through the canopy |
| g_leaf_0 | 0.25 | $[0.1, 3]$ | Minimum turnover rate for leaves (360 days)$^{-1}$ |
| dqcrit(BT) | 0.09 | $[0.045, 0.18]$ | Critical humidity deficit (kg $H_2O$ per kg air) |
| dqcrit(NT) | 0.06 | $[0.03, 0.12]$ | Critical humidity deficit (kg $H_2O$ per kg air) |
| dqcrit(C3g) | 0.051 | $[0.0255, 0.102]$ | Critical humidity deficit (kg $H_2O$ per kg air) |
| dqcrit(SH) | 0.03 | $[0.015, 0.06]$ | Critical humidity deficit (kg $H_2O$ per kg air) |
| dqcrit(Cr) | 0.075 | $[0.0375, 0.15]$ | Critical humidity deficit (kg $H_2O$ per kg air) |
| f0(BT) | 0.875 | $[0.65, 0.972]$ | Ci/Ca when dq = 0 |
| f0(NT) | 0.875 | $[0.65, 0.972]$ | Ci/Ca when dq = 0 |
| f0(C3g) | 0.931 | $[0.6916, 1.034208]$ | Ci/Ca when dq = 0 |
| f0(SH) | 0.875 | $[0.65, 0.972]$ | Ci/Ca when dq = 0 |
| f0(Cr) | 0.8 | $[0.5942857, 0.8886857]$ | Ci/Ca when dq = 0 |
| g_grow(BT) | 20 | $[10, 40]$ | Rate of leaf growth (360 days)$^{-1}$ |
| g_grow(NT) | 15 | $[7, 30]$ | Rate of leaf growth (360 days)$^{-1}$ |
| g_grow(C3g) | 20 | $[10, 40]$ | Rate of leaf growth (360 days)$^{-1}$ |

| | | | |
|---|---|---|---|
| g_grow(SH) | 30 | $[15, 60]$ | Rate of leaf growth $(360 \text{ days})^{-1}$ |
| g_grow(Cr) | 20 | $[10, 40]$ | Rate of leaf growth $(360 \text{ days})^{-1}$ |
| lai_max(BT) | 7 | $[3.5, 10]$ | Maximum leaf area index |
| lai_max(NT) | 7 | $[3.5, 10]$ | Maximum leaf area index |
| lai_max(C3g) | 3 | $[1.5, 6]$ | Maximum leaf area index |
| lai_max(SH) | 4 | $[2, 7]$ | Maximum leaf area index |
| lai_max(Cr) | 3 | $[1.5, 6]$ | Maximum leaf area index |
| nmass(BT) | 0.0257 | $[0.0089, 0.0354]$ | Top leaf N content (kgN per kgLeaf) |
| nmass(NT) | 0.01091 | $[0.00667, 0.02253]$ | Top leaf N content (kgN per kgLeaf) |
| nmass(C3g) | 0.02248 | $[0.01076, 0.05433]$ | Top leaf N content (kgN per kgLeaf) |
| nmass(SH) | 0.0192 | $[0.01, 0.0319]$ | Top leaf N content (kgN per kgLeaf) |
| nmass(Cr) | 0.0113 | $[0.00565, 0.0226]$ | Top leaf N content (kgN per kgLeaf) |
| rootd_ft(BT) | 2 | $[0.1, 5.33]$ | Parameter for decay of root functioning with depth (m) |
| rootd_ft(NT) | 1.8 | $[0.1, 4.8]$ | Parameter for decay of root functioning with depth (m) |
| rootd_ft(C3g) | 0.5 | $[0.1, 1.333]$ | Parameter for decay of root functioning with depth (m) |
| rootd_ft(SH) | 0.5 | $[0.1, 1.333]$ | Parameter for decay of root functioning with depth (m) |
| rootd_ft(Cr) | 0.5 | $[0.1, 1.333]$ | Parameter for decay of root functioning with depth (m) |
| tleaf_of(BT) | 278.15 | $[273, 283]$ | Temperature below which leaves are dropped (K) |
| tleaf_of(NT) | 233.15 | $[233, 273]$ | Temperature below which leaves are dropped (K) |
| tleaf_of(C3g) | 278.15 | $[273, 283]$ | Temperature below which leaves are dropped (K) |
| tleaf_of(SH) | 278.15 | $[273, 283]$ | Temperature below which leaves are dropped (K) |
| tleaf_of(Cr) | 278.15 | $[273, 283]$ | Temperature below which leaves are dropped (K) |
| tlow(BT) | 0 | $[-1, 1]$ | Lower temperature parameter for photosynthesis (°C) |
| tlow(NT) | 0 | $[-1, 1]$ | Lower temperature parameter for photosynthesis (°C) |
| tlow(C3g) | 10 | $[9, 11]$ | Lower temperature parameter for photosynthesis (°C) |
| tlow(SH) | 10 | $[9, 11]$ | Lower temperature parameter for photosynthesis (°C) |
| tlow(Cr) | 13 | $[11, 15]$ | Lower temperature parameter for photosynthesis (°C) |
| tupp(BT) | 32 | $[22, 36]$ | Upper temperature parameter for photosynthesis (°C) |
| tupp(NT) | 32 | $[22, 36]$ | Upper temperature parameter for photosynthesis (°C) |
| tupp(C3g) | 32 | $[22, 36]$ | Upper temperature parameter for photosynthesis (°C) |
| tupp(SH) | 40 | $[31, 46]$ | Upper temperature parameter for photosynthesis (°C) |
| tupp(Cr) | 45 | $[28, 51]$ | Upper temperature parameter for photosynthesis (°C) |
| vsl(BT) | 32.50 | $[6, 150]$ | Regression slope between $V_{cmax}$ and $N_{area}$ ($\mu$ mol $CO_2$ gN$^{-1}$ s$^{-1}$) |
| vsl(NT) | 21.46 | $[3, 96]$ | Regression slope between $V_{cmax}$ and $N_{area}$ ($\mu$ mol $CO_2$ gN$^{-1}$ s$^{-1}$) |
| vsl(C3g) | 48.03 | $[7, 322]$ | Regression slope between $V_{cmax}$ and $N_{area}$ ($\mu$ mol $CO_2$ gN$^{-1}$ s$^{-1}$) |
| vsl(SH) | 32.16 | $[15, 119]$ | Regression slope between $V_{cmax}$ and $N_{area}$ ($\mu$ mol $CO_2$ gN$^{-1}$ s$^{-1}$) |
| vsl(Cr) | 20.48 | $[9, 95]$ | Regression slope between $V_{cmax}$ and $N_{area}$ ($\mu$ mol $CO_2$ gN$^{-1}$ s$^{-1}$) |

Table 2 provides the different forcing variables that we provide the emulator. Each can take a different value for each grid cell, at each time point. Additional details can be found in Appendix C.

From here, 20,000 combinations of parameters are chosen and paired with 20,000 grid cells. This results in 9,120,000 data points in total (20,000 grid cell/parameter combinations, each for 456 8-day averages). As mentioned previously, efforts are

made to ensure that a good coverage in $\theta$ and $W_{ts}$ is obtained from these choices (by maximising a criteria discussed in Appendix B).

Running JULES for these settings then provides the data required to train the outlined emulator. Some simulations failed, and some simulations were mistakenly run for values of $f0$ which were too high. After discarding these simulations, we are left with 7,814,472 total data points which are used to train the emulator.

The resulting emulator is then used to perform some exploratory model tuning (Section 3.3) using observational data from two MODIS satellites (Running and Zhao, 2019a, b) (see Appendix D for details).

Table 2: A table showing the forcing data variables considered

| Short Form Name | Long Form Name |
| --- | --- |
| dtr | diurnal temperature range |
| huss | specific humidity |
| precip | precipitation |
| psurf | air pressure |
| rlds | longwave radiation |
| rsds | shortwave radiation |
| sfcWind | wind speed |
| tas | air temperature |
| slope | topographic slope |
| hcon1 | dry thermal conductivity in the upper soil layer |
| hcon2 | dry thermal conductivity in the lower soil layer |
| satcon1 | hydraulic conductivity at saturation in the upper soil layer |
| satcon2 | hydraulic conductivity at saturation in the lower soil layer |
| vcrit1 | critical point water content in the upper soil layer |
| vcrit2 | critical point water content in the lower soil layer |
| vsat1 | saturation point water content in the upper soil layer |
| vsat2 | saturation point water content in the lower soil layer |
| vwilt1 | wilting point water content in the upper soil layer |
| vwilt2 | wilting point water content in the lower soil layer |
| dayofyear | the day of the year |

## 3 Results

### 3.1 Emulator Performance

It is an essential part of the process to check the accuracy of an emulator, just as it is with the land surface model itself. When training the emulator(s), we hold $10\%$ of the data points aside at random. These held-out data points can then be used to test the accuracy of the emulator(s). For each of the 5 PFT-specific emulators, we obtain emulator predictions for 1000 randomly chosen points from the held-out testing data set, and obtain the 2 standard deviation intervals. These 2 standard deviation intervals should approximately correspond with the $95.4\%$ certainty interval; and so roughly $95.4\%$ of the held-out data should

**Table 3.** A table of the accuracy rates of each of the 5 pft specific emulators. A perfect emulator would have an accuracy rate of 95.4%; larger values imply under-confidence, smaller values imply over-confidence.

| GPP$_{BT}$ | GPP$_{NT}$ | GPP$_{C3g}$ | GPP$_{SH}$ | GPP$_{Cr}$ |
|---|---|---|---|---|
| 94.8% | 95.6% | 95.6% | 95.4% | 95.0% |

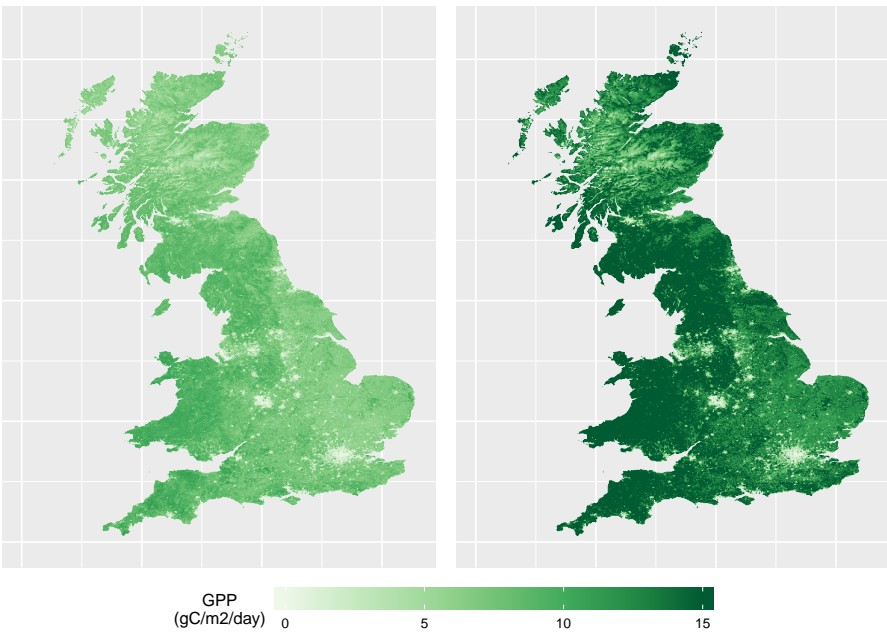

GPP
(gC/m2/day)   0        5        10       15

**Figure 2.** Two maps of GPP for June 1st 2004, as generated by the emulator, for two different parameter settings. The left plot corresponds to a non-implausible set of inputs; and the right plot corresponds to an implausible set of inputs. Both represent the emulator's mean prediction.

lie within the intervals. Table 3 shows that each of the 5 emulators performs well, with accuracy rates between 94.8% and 95.6%.

With this emulator we can then make fast predictions of 8-day average GPP without needing to run JULES again. We can do this for any time, location and scenario (whether historical or hypothetical), and for any tuning parameter settings (assuming reasonable ranges), rapidly on a personal laptop. As an example, Figure 2 shows two 1km resolution maps of predicted GPP for Great Britain, for June 1st 2004[2], obtained from the emulator, using two different tuning parameter settings.

The emulator is given no information about location, and thus the spatial structure in the predictions is inherited entirely from the forcing data provided. Other maps like these can be produced for different scenarios representing a range in environmental data, parameter settings, or PFT fractions.

The ability to emulate a land surface model at high resolution opens many potential avenues of research. For the rest of this section we explore two such avenues: sensitivity analysis and calibration.

---

[2]More accurately, as we are working with 8-day averages, these maps are for the average from May $28^{th}$ to June $4^{th}$

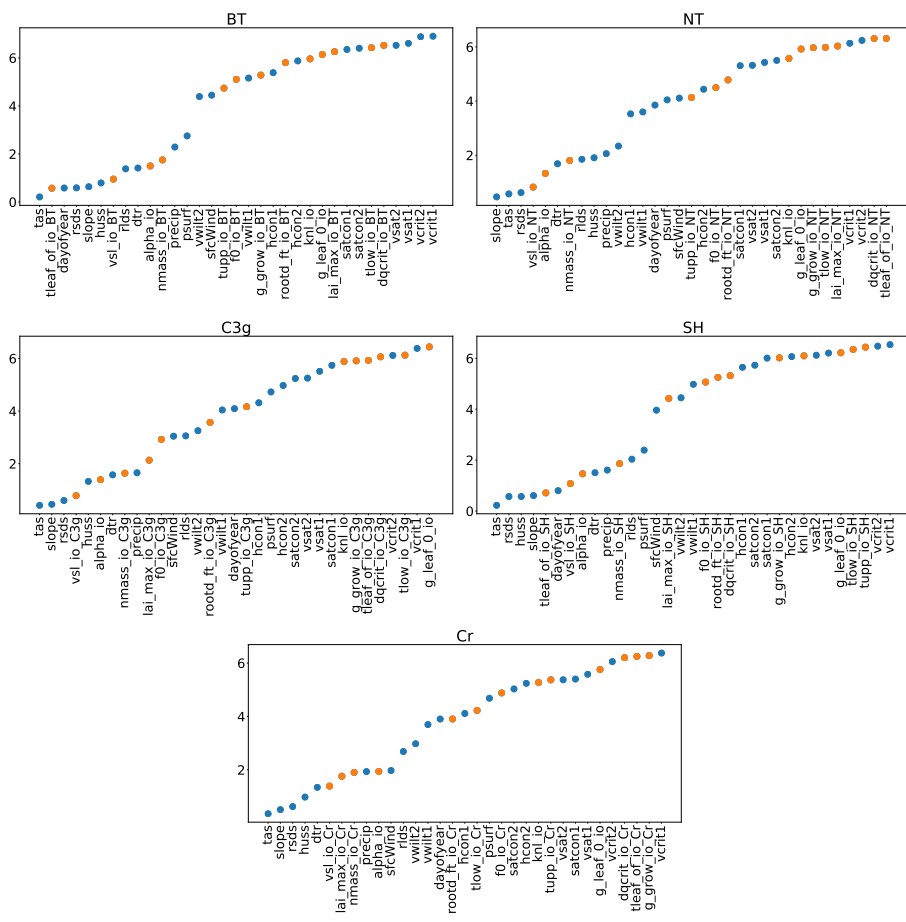

**Figure 3.** Estimated length scales from the Gaussian process emulators; a smaller value implies a stronger relationship between the input and GPP. Orange points represent tuning parameters and blue dots represent environmental data. Descriptions for the tuning parameter abbreviations can be found in Table 1, and descriptions for the environmental data can be found in Table 2.

## 3.2 Sensitivity Analysis

With a Gaussian process emulator it is possible to obtain an automatic, preliminary, sensitivity analysis. If the covariance function, $k$, in Equation (1) is chosen to be an automatic relevance determination kernel (as in this paper), the emulator will
automatically obtain estimates of the relative importance of different inputs (i.e. how sensitive the outputs are to individual inputs) if a constant mean function is also chosen (Rasmussen and Williams, 2006). This choice of $k$ is available as an option in almost all Gaussian process software, often as the default. In training the emulator, lengthscale estimates are obtained, which provide a measure of how far away two points need to be in a given dimension before they become uncorrelated. As such, a smaller lengthscale suggests a stronger relationship between the parameter and the output, and thus greater importance. Figure
3 plots these estimates for each input, for each PFT.

Clearly, each PFT has a different relationship between the inputs and GPP, but some overall patterns are visible. For one, the forcing data is, in general, more important than the parameter settings. The air surface temperature (tas), the gradient of a specific grid cell (slope), the amount of shortwave radiation (rsds), and humidity (huss) all appear important for all PFTs. For tuning parameters, the parameter "tleaf" appears important for broadleaf trees and shrubs, and the parameter "vsl" appears important for all PFTs, as does "alpha" and "nmass". "lai_max" also seems to be important for cropland and C3 grasses. Results like these can provide key insights into a LSM. More comprehensive sensitivity analysis is also possible with an emulator (Oakley and O'Hagan, 2004), with our results obtained automatically after the construction of an emulator using this covariance function.

### 3.3  Tuning/Calibration

Formal tuning, or calibration, of the various input parameters in a land surface model can also be performed more easily using an emulator. Tuning is the process of choosing the parameters such that the resulting outputs match up with real life observations. Without an emulator, an exhaustive search of the different possible parameter settings can be prohibitively expensive, and so tuning in practice often involves some degree of arbitrariness, relying heavily on subjective experience and instinct. Alternatively, an optimisation procedure can be taken (Raoult et al., 2016; Peylin et al., 2016), but this can be computationally intensive, the results will not quantify uncertainties completely, and no alternative options are provided if the final result does not agree with scientific belief.

With an emulator, many different parameter settings can be directly tested, facilitating an efficient exploration of the parameter space. History matching is a straightforward method for testing different parameter settings, ruling out inputs as 'implausible' if observed data does not match with the model output (Craig et al., 1997), and has already been successfully applied to other climate models (Williamson et al., 2013; Couvreux et al., 2021; Hourdin et al., 2021). Given observed data $\mathbf{y}_{obs}$, observational error $\sigma_{obs}^2$, a tolerance to model error $\sigma_{MD}^2$, a mean prediction for the output of the model $E[\mathbf{y}(\theta)]$ and a predictive covariance of the model output $\Sigma(\theta)$ (the latter two are provided by an emulator), the implausibility of any given parameter setting $\theta$ can be calculated as:

$$\mathcal{I}(\theta) = (\mathbf{y}_{obs} - E[\mathbf{y}(\theta)])^\top \left( \Sigma(\theta) + \sigma_{obs}^2 I + \sigma_{MD}^2 I \right)^{-1} (\mathbf{y}_{obs} - E[\mathbf{y}(\theta)]), \tag{6}$$

where $I$ is the identity matrix. Implausibility is similar to mean square error, but each grid cell is weighted according to its uncertainty (which has components due to observation error, structural model error and emulator variance). A larger implausibility indicates a greater mismatch between the observations and the output from the land surface model (indicating that the parameter setting, $\theta$, can be ruled out). A conservative threshold for rejecting a parameter setting can be taken as the $99.5\%$ quantile of the $\chi^2$ distribution with $l$ degrees of freedom (where $l$ is the dimension of the observation) (Vernon et al., 2010).

To demonstrate history matching for JULES, we consider a small subset of our grid cells (1000 points chosen to maintain a good coverage of the forcing data), and we randomly sample a time point for each coordinate. We then use observed GPP data from MODIS (details in Appendix D) for these locations and times, taking the mean of two separate observations as

'the' observation. We take the observational error standard deviation $\sigma_{obs}$ as the standard deviation between the two individual observations plus an additional, conservative, $20\%$ of the mean value (because a standard deviation estimate obtained from only 2 samples will be very inaccurate).

Tolerance to model error (model discrepancy) is an inherently subjective concept, and it is almost impossible to learn this from data when there are also uncertain tuning parameters, at least not without strong prior beliefs (Brynjarsdóttir and O'Hagan, 2014). To elicit subjective values for the model discrepancy variance, we use the three-point Pearson and Tukey formula outlined in Keefer and Bodily (1983) and Revie et al. (2010). With this, we need to provide several values quantifying how flawed we believe JULES could be. Specifically, we have to provide the $5\%$, $50\%$, and $95\%$ quantiles for what we believe the difference between real GPP and the estimates of GPP from JULES would be, even if we obtained the "correct" parameter values. For this, we assume the difference could be up to $40\%$ of the real GPP value (our $95\%$ quantile), or down to $20\%$ of the real GPP value (our $5\%$ quantile), whilst assuming the output from JULES is still the best guess for what the real GPP could be (0, our $50\%$ quantile). With these beliefs, the Pearson and Tukey formula imposes a mild bias (as our hypothetical distribution for the failings of JULES is skewed), which needs to be added to the $(\mathbf{y}_{obs} - E[\mathbf{y}(\theta)])$ terms in Equation (6). To calculate these values, we use the observed data as a substitute for the real GPP values. Because of how we quantified our model discrepancy beliefs, our values for the model discrepancy bias and variance varies from grid cell to grid cell.

We then consider 100,000 different candidate parameter settings ($\theta$, selected using a maximin latin hypercube to ensure a good variety (McKay et al., 1979)); and immediately rule-out roughly $85.5\%$ as implausible, leaving 14,475 non-implausible parameter settings.

Some parameter settings are not ruled-out because they result in sufficiently accurate GPP predictions. Some are not ruled-out because the uncertainty from the emulator is too large. To rectify this, more JULES runs can be made for parameter settings which have not yet been ruled-out, improving the confidence of the emulator in the non-implausible regions of parameter space, and the tuning process can then be repeated. This is common practice with history matching, and is called "iterative refocussing" (Hourdin et al., 2021). We obtain another 'wave' of 3000 JULES runs, using 3000 different non-implausible parameter settings. These parameter settings are chosen as a subset of the 14,475 non-implausible parameter settings found, with the subset chosen using the same technique used to choose the initially simulated *grid cell* subset in Section 2.4. The grid cells for the new runs were made up of 2000 new coordinates chosen via the same technique in Section 2.4, and the 1000 coordinates that we have observational data for. These parameter settings and grid cells are then combined via the same technique for matching parameter settings and grid cells used in Section 2.4.

The combined set of the new JULES runs and the previous runs for parameter settings which are not yet ruled-out makes up the second wave's dataset. We fit a new emulator using the same procedure as before, which is then used to further rule-out parameter settings. We find a further $69.9\%$ of parameter settings to be implausible; leaving a total of 4363 non-implausibile parameter settings, or $4.4\%$ of the originally considered set of parameter settings. This procedure could be repeated further, until no changes are found. At any stage, the non-implausible parameter settings represent the parameter choices which sufficiently agree with the observed data and simulations, considering the various uncertainties.

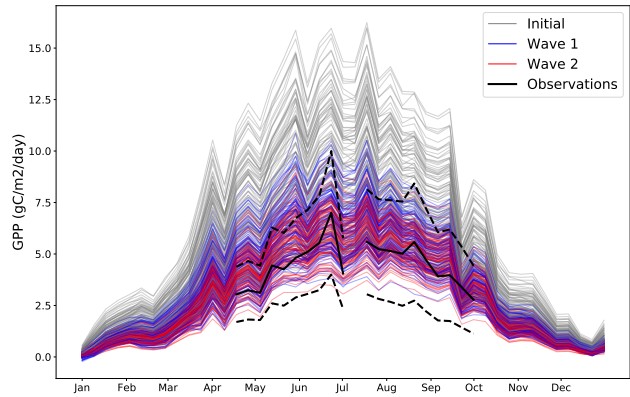

**Figure 4.** Different time series of GPP for the year 2002 for a randomly chosen coordinate where we have observations. Grey lines represent 100 randomly chosen parameter settings before any tuning is performed. Blue lines represent 100 randomly chosen parameter settings after the first wave of tuning. Red lines represent 100 randomly chosen parameter settings after the second wave of tuning. The black solid line represents the observations for that coordinate (and the black dashed lines represent the 2 standard deviation intervals of the observational error). By its nature, all possible wave 2 trajectories must be a subset of all possible wave 1 trajectories, which in turn must also be a subset of all possible initial trajectories.

Interestingly, we found 'canonical JULES' (i.e., the set of standard values considered) to be non-implausible in the first wave, but with more simulations, we found it be implausible in the second. This suggests that improvements in the standard settings can be made.

The map on the left in Figure 2 was produced using a non-implausible parameter setting, and the map on the right used an implausible parameter setting. There is a clear difference between the two maps. Spatial gradients in the not ruled-out yet parameter setting are much more gentle; with no large extremes in GPP, but still a distinct spatial pattern (for example, lower GPP in the Scottish highlands, and a decrease in GPP going from west to east and corresponding to rainfall gradients). The GPP map corresponding to the ruled-out parameter setting appears very extreme; predicting larger changes over relatively small areas, as well as generally very large values for GPP, with the extremes $> 15gC/m^2/day$[3].

As an illustrative example of the iterative tuning process; Figure 4 presents different possible trajectories from different possible parameter settings.

This plot shows how the distribution of possible trajectories of the model output change after different waves of tuning. Initially, the range of possible trajectories extends far beyond what is possible according to the observations. By wave 2, the trajectories closely follow the observations, with their allowed spread appearing to be determined entirely by the observational error. The improvement between wave 1 and wave 2 is visible, but minor, suggesting that an additional wave of simulations is unlikely to be worthwhile.

---

[3]$gC/m^2/day$ is used as the unit here, but these GPP predictions are still for the 8-day average

## 4  Discussion

In this work, we have outlined a framework for emulating land surface models using sparse Gaussian processes. This framework takes into account a unique feature of many land surface models, incorporating the information contained within the forcing data. Under this framework, a substantially better coverage of the input parameters can be obtained in the initial simulation ensembles, without additional simulation effort and without compromising analytical capabilities. The use of sparse GPs for emulating a land surface model is the first we are aware of and is natural here where the ensembles are made artificially large, relative to standard computer experiments. Specific modifications were made to build an emulator for JULES, but the overall procedure can remain the same for many LSMs. However, every land surface model is different, and so various modifications to the procedure should be made depending on the specific priorities and interests of the modeller.

Such an emulator can be used to achieve many experimental goals. Being able to obtain fast predictions for new simulations can unlock comprehensive analysis and exploration of the various relationships within an LSM, even at high resolution. Any goal that would be better achieved with unlimited simulation from the LSM could benefit from an emulator acting as a fast surrogate model. Sensitivity analysis and calibration of input parameters are two potential applications which were explored briefly in this work. A promising potential application is optimisation; where one tries to maximise (or minimise) an output of the land surface model (or a metric that can be created from various outputs). The optimisation of a system using a high-resolution land surface emulator could be an effective tool in informing local policymaking regarding changing land use. Optimisation usually requires many evaluations of the process in question, and local policymaking will require expensive high resolution evaluations, and so an emulator like the one outlined here is a valuable asset. Optimisation using Gaussian processes is a well-researched topic (Mockus, 2012; Jones et al., 1998), and is often referred to as "Bayesian Optimisation".

Along with extensions to practical use, there are a few methodological extensions to this work that can be envisioned. With our emulator framework, model predictions should be valid for any future or alternative scenario, assuming the driving data for said scenario does not exceed the extremes of the driving data used to train the emulator. If the driving data for a new scenario goes beyond the extremes from the training data, then the emulator will begin to extrapolate, and accuracy will decrease when and where the training data extremes are exceeded. With a changing climate, this is more likely to occur (although one would hope that the driving data, such as the air temperature, will not soon constantly exceed the highest value observed in the historical training record). This potential problem could be avoided by training a land surface emulator using partially artificial driving data, along with observed driving data as we have, with the artificial driving data providing coverage for potential extreme, unseen, scenarios.

Also, for the outlined emulator framework, the modification wherein the high dimensional spatial-temporal output is converted to a large set of 1D outputs allows the information in the forcing data to be readily incorporated and the dimension of the output to be shrunk. However, removing the time structure (instead of only the spatial) is, in many cases, a simplification rather than an assumption. Re-adding the time structure may be an interesting direction for future work. This could be via a dynamic emulator as discussed previously, by using dimension reduction techniques over the time dimension, or by including previous time steps of the driving data $W_{(t-i)s}$ as inputs to the emulator. The current modifications explained in Section 2.2

seem to provide an acceptably accurate emulator. Although the 95% certainty intervals produced by the PFT-specific emulators for JULES contain the truth roughly 95% of the time, the emulators can still be considerably erroneous on rare occasions. Also, even though GPP is always greater than 0, sometimes the emulators will predict GPP to be less than 0 (which is easily rectified by converting any negative predictions to 0, but this does distort the uncertainty intervals when GPP is predicted small/negative). As such, it should be noted that, just like a land surface model, an emulator can always be improved upon.

Regarding JULES itself, the preliminary sensitivity analysis in Section 3.2 identified that, in general, the forcing data has a greater influence on GPP than the parameter settings. This is perhaps obvious, as the amount of plant activity depends heavily on the environment; but this does suggest improving the accuracy of environmental data should be a key priority for practitioners working with JULES. This agrees with the results from McNeall et al. (2020). The sensitivity analysis performed in Section 3.2 was only a preliminary, automatic, sensitivity analysis. A more comprehensive sensitivity analysis using this emulator (perhaps in the format of Oakley and O'Hagan (2004)) is left for future work.

For the tuning procedure, more waves of simulations could be added until no greater improvements could be made. Alternatively, one non-implausible parameter setting could be hand-selected via expert opinion. Additionally, the observational data used was of relatively poor quality, with a large observational error that would have hindered the rejection of some poor parameter choices. All low values of GPP (<2 gC/m2/day) were also excluded from the observational data because of a mismatch between the two sources of data (Appendix D), which means that many parameter choices which poorly recreate low GPP may not have been rejected. Better observational data, with lower observational uncertainty and requiring fewer deletions / with less missing data, would further shrink the parameter space, improving the tuning results.

Similarly, while the obtained tuning for JULES ensured that the remaining parameter settings match well with observed GPP, this does not necessarily mean that the parameters will match well with other outputs of JULES. Emulating and matching multiple distinct outputs is an interesting avenue for future experimentation with JULES, especially when it is entirely possible that certain parameter settings will be good for one output, but mutually exclusive parameter settings will be good for a different output (this effect is observed by McNeall et al. (2020)). This would mean that no parameter settings will provide an overall good match (the so-called terminal case (Salter et al., 2019)), and suggests the degree of model discrepancy is larger than initially thought. Discovering which observations a land surface model can and cannot reproduce, and which parameter settings are better and which are worse, can be useful information for quantifying and rectifying the flaws in a land surface model.

As an additional note regarding tuning: reducing the set of non-implausible parameter settings does not necessarily impose any individual bounds on the various parameters. For example, if there were only 2 parameters, both promoting plant growth, then it is entirely reasonable to believe a large value for the first and a small value for the second could match with reality, while believing that a small value in the first and a large in the second could also match with reality. What would be important here, is to rule-out small values for both and large values for both. With 53 parameters, rather than just 2, this situation is essentially guaranteed.

A related technique to tuning is that of data assimilation (Reich and Cotter, 2015), which is common in the wider environmental science community; the best known application being weather forecasting. With data assimilation, at each timestep

within a specific simulation, the simulation results (the state variables) are adjusted to match more closely to the observed data, before resolving the next timestep. This goal is somewhat related to tuning, because both can result in better matched simulations and observations, but the way in which this is done and the overall experimental objectives differ. One could tune a model's parameters but then also perform data assimilation on a final simulation. Data assimilation methodology can also be used to simultaneously tune model parameters and adjust the model state, avoiding the need for a specialised tuning procedure, but it is not clear if such a strategy is effective (Rougier, 2013).

As a reminder, a key benefit of an emulator is its computational speed compared to running the full model. For comparison, simulations of JULES over Great Britain on a 1.5 km × 1.5 km resolution can take approximately 9.5 hours per decade for the spin up (a warm-up period where detailed outputs are not produced) and approximately 25.5 hours per decade in similar model runs to the ones used in this paper. These were run on 72 processors on the Met Office and NERC Supercomputer Node phase 2 (MONSooN2). The emulator however, on a higher 1km × 1km resolution, for Great Britain, takes roughly 20 seconds on a personal laptop for an 8-day prediction using 1 core. This would correspond to approximately 2.5 hours per decade. There are a few important caveats to this which can make the emulator even faster in practice. Firstly, the emulator predictions are fully parallelizable, and so speed increases are easily achieved through using more cores. Secondly, these emulator prediction times are for a personal laptop, which we would expect to perform worse than a supercomputer. Thirdly, sparse Gaussian processes still scale with the number of training points used, and the 9,120,000 data points used for this emulator is probably excessive by at least one order of magnitude. Fourthly, and perhaps most importantly, the emulator doesn't need spin-up time, and doesn't need to be run for a whole time-series. This makes it easier to directly obtain the desired information, for the desired times and desired locations; long run periods are not always needed. For example, the tuning performed in Section 3.3 did not need the entirety of Great Britain for the entire historical record; and instead the emulator was able to explore many different values of the tuning parameters for a select few times and locations - something which would not be possible without an emulator.

As a final comment from this work, hopefully we have highlighted the relative accessibility of modern statistical techniques. Gaussian process emulators are sophisticated, but they can also be intuitive, whilst also being incredibly capable. Software for implementing Gaussian process emulators is readily available and abundant. Compared to the expertise required to initially develop a complex land surface model, relatively basic expertise is required to extract substantially more value from the model using an emulator.

## Appendix A: Sparse Gaussian Processes

What follows is a technical overview for sparse Gaussian processes and how they are fit. Full discussion and derivation is left to the cited references. Implementation for sparse Gaussian processes is readily accessible via publicly available software such as the python package `GPflow`.

Consider a mean zero Gaussian process (as used throughout this article)[4], with kernel $K(\cdot,\cdot)$. An alternative way to express the GP formulation for an ensemble of model output, $\boldsymbol{y}$, is

$$\boldsymbol{y}\mid\boldsymbol{f},\sigma^2 \sim \mathrm{N}(\boldsymbol{f},\sigma^2 I) \tag{A1}$$

$$\boldsymbol{f} \sim \mathrm{N}(\boldsymbol{0},K_{nn}) \tag{A2}$$

where $K_{nn}$ is the matrix $k(X,X)$. This formulation says that the model data is an observation of an underlying GP, $\boldsymbol{f}$, and will allow for computational simplifications.

Posterior inference requires the inversion of $K_{nn}$, which becomes impractical if $n$ is large, and is particularly problematic for fully Bayesian inference (which requires this inversion for every proposed parameter sample) but also for methods where the parameters are optimised and plugged in.

A solution to this lies in a variational approximation to posterior inference based on augmenting $\boldsymbol{f}$ with $m << n$ pseudo-simulations, $\boldsymbol{u}$, at input locations $Z$, called "inducing points". Variational methods propose an approximate parametric posterior distribution, $q(\cdot)$, known as the variational posterior, and find the parameters for $q()$ which minimise the Kullback-Leibler divergence between $q()$ and the true posterior. This is equivalent to maximising a well known lower bound on the marginal likelihood (Salimbeni and Deisenroth, 2017):

$$L = \mathrm{E}_{q(\boldsymbol{f},\boldsymbol{u})}\left[\log\frac{p(\boldsymbol{y},\boldsymbol{f},\boldsymbol{u})}{q(\boldsymbol{f},\boldsymbol{u})}\right]. \tag{A3}$$

Taking the variational posterior as $q(\boldsymbol{f},\boldsymbol{u}) = p(\boldsymbol{f}\mid\boldsymbol{u})q(\boldsymbol{u})$, with $q(\boldsymbol{u}) = \mathcal{N}(\boldsymbol{u};\boldsymbol{m},\boldsymbol{S})$ (where the slight abuse of notation, $\mathcal{N}(\boldsymbol{u};\boldsymbol{m},\boldsymbol{S})$, is used to represent the Normal density for $\boldsymbol{u}$ with mean $\boldsymbol{m}$ and variance $\boldsymbol{S}$) leads to $p(\boldsymbol{f}\mid\boldsymbol{u})$ cancelling in Equation (A3), because $p(\boldsymbol{y},\boldsymbol{f},\boldsymbol{u}) = p(\boldsymbol{y}\mid\boldsymbol{f})p(\boldsymbol{f}\mid\boldsymbol{u})p(\boldsymbol{u})$:

$$L = \mathrm{E}_{q(\boldsymbol{f},\boldsymbol{u})}\left[\log\frac{p(\boldsymbol{y}\mid\boldsymbol{f})p(\boldsymbol{u})}{q(\boldsymbol{u})}\right].$$

Additionally, $\boldsymbol{u}$ can be analytically integrated out from $q(\boldsymbol{f},\boldsymbol{u})$ (required for the expectation) to leave:

$$q(\boldsymbol{f}\mid\boldsymbol{m},\boldsymbol{S}) = \mathcal{N}(\boldsymbol{f}\mid\mu,\Sigma),$$

with

$$\mu = K_{nm}K_{mm}^{-1}\boldsymbol{m}$$

$$\Sigma = K_{nn} - K_{nm}K_{mm}^{-1}(K_{mm}-\boldsymbol{S})K_{mm}^{-1}K_{nm}^{T},$$

where $K_{mm}$ is the matrix $k(Z,Z)$, and $K_{nm}$ is the matrix $k(X,Z)$. This formulation only requires the inversion of $m \times m$ matrices, providing a significant computational speed-up.

Collecting the terms of $L$ that depend on $\boldsymbol{u}$ gives

$$L = \mathrm{E}_{q(\boldsymbol{f}\mid\boldsymbol{m},\boldsymbol{S})}\left[\log p(\boldsymbol{y}\mid\boldsymbol{f})\right] + \mathrm{E}_{q(\boldsymbol{u})}\left[\log\frac{p(\boldsymbol{u})}{q(\boldsymbol{u})}\right], \tag{A4}$$

---

[4]This is a modelling choice that is common with some authors (Binois et al., 2018), whilst more complex mean functions are favoured by others (Bower et al., 2010). A discussion of these choices can be found in Section 2.7 of Rasmussen and Williams (2006)

where the second term is the KL divergence between prior $p(u)$ and $q(u)$. The first term can be expressed as a sum of univariate expectations of individual data points so that

$$L = \sum_{i=1}^{n} \mathrm{E}_{q(f_i|\boldsymbol{m},\boldsymbol{S})} \left[ \log p(y_i \mid f_i) \right] + \mathrm{E}_{q(\boldsymbol{u})} \left[ \log \frac{p(\boldsymbol{u})}{q(\boldsymbol{u})} \right].$$

As such, during optimisation of $L$ for the various parameters ($\boldsymbol{m}, \boldsymbol{S}$, $Z$ and the covariance hyperparameters), the data can be sub-sampled at each iteration (still obtaining an unbiased estimator for $L$). These smaller sub-samples are called minibatches,
and provide a second computational speed-up (Hensman et al., 2013).

## Appendix B: Choosing JULES runs

Throughout this article we mentioned how the various JULES simulations were chosen to maintain a good coverage of the forcing data $W_{ts}$ and the tuning parameters $\theta$. The sheer number of runs that were possible under this framework reduces the importance of carefully selected simulations, but we will outline our method here for completeness.
In JULES, because outputs at time $t$ are dependent on state variables from time $t-1$, a grid cell must be run for the entire time range of interest (rather than only for the time steps required). As such, we could not simply select $W_{ts}$ to have good coverage, we instead had to select a set of grid cells such that the resulting set of $W_{ts}$ had good coverage. To do so we condensed the time series $\mathbf{w}_t$ for a given grid cell $s$ into it's historical mean $\hat{\mathbf{w}}_s$ from 2000-2009, and ensured a good coverage of these means were obtained.
To ensure a good coverage was obtained, we used the combination of three distinct scores. The first score is the minimum distance between any two $\hat{\mathbf{w}}_s$ within a potential set of these historical means, $\hat{W}_s$. The distance between any two $\hat{\mathbf{w}}_s$ represents how similar any two grid cells are, and so maximising this minimum distance $M$ ensures no two chosen grid cells are too similar.

The second score is based on Latin hypercubes. Latin hypercubes are designs where, after binning each dimension $d_i$, each
bin in each dimension is forced to contain only one point (McKay et al., 1979). A Latin hypercube ensures that every dimension has good coverage. As we are not designing our grid cells from scratch, and instead have a large set to choose from (those that exist in Great Britain), obtaining an actual Latin hypercube is most likely impossible. Instead, we score a potential set of $\hat{W}_s$ based on how close to a Latin hypercube it is. Our Latin hypercube score is:

$$L = \sum_{d_i} \sum_b |P_b - A_b| \tag{B1}$$

where $b$ is the index for a bin, $P_b$ is the preferred number of points in the bin, and $A_b$ is the actual number of points in the bin. In a real Latin hypercube, because there would be as many bins $b$ as desired points in the ensemble, $P_b$ would always equal 1, and $A_b$ would always equal 1. As such, for a real Latin hypercube, $L = 0$. The further away $\hat{W}_s$ is from 0, the less like a Latin hypercube it is. Because we are dealing with a large number of desired data points, we set the number of bins in each dimension equal to 20, and so $P_b$ is equal to $n/20$ (where $n$ is the number of desired ensemble members).

These two scores together constitute a type of 'maximin Latin hypercube score'. Normally this would be sufficient, but as mentioned before, we have had to work with the historical means $\hat{W}_s$ rather than the actual inputs $W_{ts}$. The intra-cell variation does still matter for the emulator, as a grid cell which has a very variable climate is more useful - providing more information into how JULES reacts to different forcing values. As such, our third score, $C$, is simply the difference between the historical $90^{th}$ percentile and the historical $10^{th}$ percentile (summed over each grid cell in the potential set of grid cells $\hat{W}_{ts}$, summed

over each forcing dimension) - providing a measure of how variable that grid cell is[5].

To select a set of grid cells to run JULES for, we then combine these individual scores as:[6]

$$\lambda_M M - \lambda_L L + \lambda_C C. \tag{B2}$$

A set of grid cells with a higher combined score is a better set of grid cells to run. We randomly obtain many potential sets of grid cells and score each of them, the highest scoring set of grid cells is then the one chosen. The $\lambda_i$ in this equation are

weights to ensure the three component scores are on the same scale, and equal $1/(max(i) - min(i))$.

To choose the set of parameters that go with the grid cells, many Latin hypercubes (of dimension 53) were generated (which is possible because the tuning parameters $\theta$ can take any value in their domain), and the combined design $(\hat{W}_s, \theta)$ with the maximum minimum distance was chosen.

More potential designs can be obtained if these schemes are run for a longer period of time, thus providing a better final

result. The wave 1 grid cell selection scheme, the wave 1 $\theta$ selection scheme and the calibration data grid cell selection scheme were all ran for 5 hours. For the wave 2 design the grid cell selection scheme and the $\theta$ selection schemes were all ran for 1 hour. These times are certainly excessive, especially given the large number of JULES runs obtained, and a much shorter run time would have been sufficient.

**Appendix C:  JULES Configuration**

The JULES configuration used in this study is JULES version 5.6, based off Blyth et al. (2019). Full details are provided in that paper, but here we provide relevant background information for interpreting the emulator results. We also note any changes from Blyth et al. (2019). The 5 PFTs used were chosen for their relevance to UK landscapes. Three non-vegetated surfaces can also be present in each grid cell: lakes, urban, and bare soil. The fractional coverage of these eight tiles is prescribed in each grid cell based on the CEH Land Cover Map 2000 (Fuller et al., 2002). The soil is represented by four discrete layers (0.0-0.1

m, 0.1-0.35 m, 0.35-1.0 m, and 1.0-3.0 m), with the Van Genuchten approach used for calculating soil hydrology. The relevant parameters are calculated from textures taken from the Harmonised World Soil Database (Nachtergaele et al., 2012). These parameters are: dry thermal conductivity (hcon), hydraulic conductivity at saturation (satcon), volumetric water content at the wilting point, a critical point (below which GPP is reduced due to soil moisture stress), and the saturation point (vwilt, vcrit, and vsat, respectively). In this work, each layer is treated as a different input to the emulator, and only the top and bottom layers

---

[5]Note that, for $M$, $L$, and $C$ here, the forcing data dimensions are all scaled to be between 0 and 1, ensuring a fair balance for each dimension

[6]When using this equation to choose a subset of non-implausible parameter settings to use in the second wave of history matching, the $\lambda_C C$ term is excluded (as it does not exist for parameter settings).

are considered, as the central two mimicked the outer two. Topographic slope is also used to calculate saturation excess runoff; this is derived from a 50 m resolution database - the CEH Institute of Hydrology Digital Terrain Model (IHDTM) (Morris and Flavin, 1990, 1994).

The driving meteorology, described in Robinson et al. (2017), is a combination of gridded daily precipitation observations and other meteorological observations. The original data resolution was 40 km, and this was downscaled to 1 km based on 510 topography (Blyth et al., 2019). There are eight meteorological driving variables: the diurnal temperature range (dtr), air temperature (tas) specific humidity (huss), precipitation (precip), air pressure (psurf), longwave radiation (rlds), shortwave radiation (rsds), and wind speed (sfcWind). We take values for these forcing variables from the CHESS data set (Robinson et al., 2017).

The model is run at a half-hour time step. The daily driving meteorology is interpolated to half-hourly using a disaggregation 515 scheme (Blyth et al., 2019). Model fluxes such as GPP, NPP, respiration, latent heat, sensible heat, and runoff are calculated every half hour. At the end of each day, a phenology scheme updates leaf area index based on temperature functions of leaf growth and senescence. Every ten days, the dynamic vegetation component of JULES (TRIFFID) updates vegetation and soil carbon stores. Competition between vegetation types is turned off in this configuration of the model. Note that in Blyth et al. (2019), LAI was prescribed rather than predicted, and TRIFFID was not used to update carbon stores.

**Appendix D: Tuning Data**

In the tuning procedure, we used the MOD17A2HGF and MYD17A2HGF Version 6 gap-filled GPP data as a test for model tuning and calibration (Running and Zhao, 2019a, b). The observed GPP data is a cumulative 8-day composite of values within 500 m, from the Terra (MOD17A2HGF) and Aqua (MYD17A2HGF) satellites. The data was downloaded from the NASA Application for Extracting and Exploring Analysis Ready Samples (AppEEARS, AppEEARS Team. (2020)). The dataset 525 comes with quality flags, and we only included 'Good quality' data with clear skies and where the main (RT) method was used, yielding either 'best result possible' or 'good very usable' result. We extracted the 500 m pixel closest to the center of the CHESS grid cell used in the JULES simulation. When both Terra and Aqua had 'good' retrievals for a given time and location, we used an average of the two. An additional constraint was necessary, as sometimes one satellite retrieved a moderately high GPP while the other retrieved a distinctly small value. As a conservative fix, we ignored both retrievals when one retrieved a 530 value $< 2$ gC/m2/day.

*Code and data availability.* Both the JULES model code and the files for running it are available from the Met Office Science Repository Service: https://code.metoffice.gov.uk/. Registration is required and code is freely available subject to completion of a software licence. The JULES outputs were obtained from running JULES version 5.6: https://code.metoffice.gov.uk/trac/jules/browser/main/trunk?rev=15927. The runs were completed with the Rose suite u-bo065: https://code.metoffice.gov.uk/trac/roses-u/browser/b/o/0/6/5/trunk

Simulations used to fit the emulator are available at https://doi.org/10.5285/789bea37-0450-4822-9857-3dc848feb937. Python code to build and validate the emulator is provided in the supplementary material.

*Author contributions.* EB developed the emulator, performed the analyses, and wrote much of the article. AH prepared and ran the simulations, provided insight into JULES, obtained the observational data, and edited the article. DW aided in the development of the emulator, and edited the article. PC aided in the development of the emulator and the general scientific discussion of the problem.

*Competing interests.* The authors declare that they have no conflict of interest.

*Acknowledgements.* This work was supported by the Natural Environment Research Council [NE/T004177/1].

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
