# Peer review of "Emulation of high-resolution land surface models using sparse Gaussian processes with application to JULES"

_Geoscientific Model Development, 2021_

## Author Response (AR1)

**Reviewer 1:**

Thank you for your comments, they were very thorough and found a lot of genuinely good corrections. We have made all the suggested fixes. Below is a (long) set of specific replies, most of which are just affirming the correction has been done.

- Line 8 (in abstract): '...acknowledges the forcing data...' Would a general reader of the abstract know what is meant by the term 'forcing data' here? I think it would be useful to clarify this.

We now give a brief explanation for what we mean by forcing data in the abstract. (on what is now lines 9-10)

- Line 28-29: '...(with 20 years of spin up simulation... took 1 additional hour)...' This might be a naïve question, but I find this is a little confusing – why does 20 years of spin-up simulation only take one hour, when it takes 30 hours to simulate 11 years of simulation?

Excellent catch. It took roughly 1 hour per year, not one hour in total. We've now fixed this. (this is still a little faster than the actual simulation, but much more reasonable. Spin up can be quicker (for example, because outputs don't need to be saved), but that much faster! The numbers before were also rounded numbers, and so we've updated them to be more accurate. (on what is now lines 31-32).

- Line 58: '... these methods ignore spatial and temporal dependence, but rely on speed...' I don't think this quite tells the full story for these methods – they do ignore spatial and temporal dependence in the emulator models, but under the assumption that spatial/temporal dependence carries through to a reasonable extent via the training data that is used to construct the emulators. Please edit the sentence here to reflect this.

We have added to this sentence to try and not undersell the value of these methods. (on what is now lines 60-62)

- Line 89: 'standardise the outputs, and this is also done in this work'. There is no detail on how outputs are standardised in the application later on in the paper? For reproducibility, please clarify how this is done.

We have clarified what we mean by "standardise" (on what is now lines 102-103)

- Line 103: Is the vector of forcing variables,  $w_{ts}$ , just being treated like extra parameters for the emulator fit here? Please make this clear in the text. [I think it is, but it took me a while to realise this.]

Yes, you are right. We have made this clear in the text. (on what is now line 118)

- Line 111: 'working with 8-day averages'. Is the input data for the 'forcing variables' also converted to 8-day averages to match the output y? - So that the temporal nature of both is on the same resolution for the emulator?

Yes, we state this explicitly now. (on what is now line 126)

- Line 112: '...we supplement the forcing data  $W_{ts}$  with a pseudo-forcing variable that records the day of the year...' I don't understand this. How is this done? How does this 'pseudo forcing variable' link to the 8-day average? [Where is this day in the 8-day average? Day 1? Centred (4 or 5)?...] How is it used in the application? (It isn't mentioned in Sect 2.5.) More detail would be useful.

Day of the year is included in  $W_{ts}$  as an additional column, treated the same as any other. In that way, day of the year is like another input parameter, just like precipitation or air temperature. For the specific value, we used the 5th day in the 8 day average, although the specific choice is not important, as long as it is consistent.

We have tried to be more clear about both now in the text. (on what is now lines 128-130)

- Section 2.3, Lines 117-121: This paragraph seems to imply (via the use of the term 'data point' in the sentences) that the 'n' data points of the n x n covariance matrix inversion for a Gaussian Process in Section 2.2 (line 121) is of the same magnitude as the number (tens of millions) of data points that a single simulation produces across the full grid (line 119). Is that true? Please clarify the relative magnitude of 'n' for the matrix inversion that is encountered for the application.

Yes, tens of millions would indeed be the 'n'. We do a trick later to bring that down to a slightly smaller number (millions instead, which we now explicitly include as well, and is a later correction), but yes, tens of millions is indeed the sort of order of magnitude of 'n' here. We've tried to make this a bit clearer now. (on what is now line 142)

- Lines 129 / 134: Are the inducing points Z used for the application a subset of the training data points X from the JULES simulations that are run, or can they be at  $(W_{ts}, \theta)$  combinations that are outside the training set?

*They can indeed be anywhere in the input space, and we mention that explicitly now. (on what is now lines 149-150)*

- Line 133: '...during each iteration of the inference.'. It is not clear to me how the inference is actually carried out? Through an MCMC-type procedure? What is involved in an 'iteration of the inference' when fitting the emulator? Please can you supply a little more information to make it easier to understand how the method is actually applied? [What is GPflow doing?]

Inference here is optimisation of the variational lower bound, to find the values of the inducing points and the GP hyperparameters.

We've tried to be more clear that the process is one of optimisation. (on lines 151 – 156). Further details are left to the appendix, to improve the flow for non-statistician readers.

**- Line 143: I think it would be very useful to reference the Appendix B here, and connect your approach to selecting the training data for your application in to this section (2.4).**

We have added reference to Appendix B here, and slightly fleshed out the description here (on what is now lines 163-165)

- Line 154: Can you also state the number of grid-cells in total that cover Great Britain at this resolution here? – to make clearer the full extent of the spatial grid being covered.

We have added a footnote on page 7 that mentions the number of grid cells.

Section 2.5: Please can you add more information to this section to explicitly describe what the 'forcing data' for the model actually is? How many variables? What are they? Maybe list them in a table format like Table 1? I think this is important to the application, especially with the sensitivity plots in Section 3.2 showing that many of these environmental variables are highly important (more important than the uncertain parameters), yet there is no description of it until very briefly when you get to the end of Appendix C, which is a bit late.

**A table has been added listing the different forcing variables on what is now page 10.**

- Line 174: '20,000 combinations of parameters are chosen and paired with 20,000 grid cells'. How many training 'data points' does this actually lead to for fitting the emulators? Stating this clearly will give the reader a full appreciation of the amount of training data you have and so a better understanding of quantities such as '10% of the data points' on line 181...

*Text has been added explicitly mentioning the number of data points that we have (on what is now lines 202-208)*

- Line 190-191: '... shows ...maps of predicted GPP for Great Britain, for June 1st 2004, obtained from the emulator,...'. How do you get a map of the output (GPP) for a specific day, when the GPP output used for the emulator is an 8-day average? More detail/clarity on how the emulator is sampled to produce the map is needed.

Good spot. More accurately, these are the average for May  $28^{th}$  – June  $4^{th}$ . A footnote has been added to page 11 stating this.

- Line 198-203: Are these the choices of the covariance function k and mean function m for the fitted emulators? It would be very useful for re-producibility if these choices were clearly stated somewhere in the Methods section (Section 2).

Added explicit mention of which mean and covariance functions were used to what is now lines 99-100.

- Line 205: Should 'each input' at the end of this sentence actually be 'each PFT'?

Should probably have said "for each input, for each PFT", which is what we've now changed it to say (on what is now line 235)

- Line 238-243: 'We obtain a value for the tolerance to model error...' I don't understand fully what is done here – This needs more explanation. The formula in the references looks to estimate the 5th, 50th and 95th percentiles of a distribution on the discrepancy term – Are the '20% smaller' and '40% larger' equated to those 5th and 95th percentile points? What is used for the 50th percentile point? - Is that coming from the model (JULES, or the emulator?) or the MODIS observations? Is a different value of the tolerance to model error (and so different 'mild bias') obtained for each observed grid-cell? How is the actual value of the 'mild bias' obtained?

We've tried to make it clearer what's happening here, roughly doubling the amount of text for this explanation. To summarise, model discrepancy (sometimes called tolerance to model error) is an inherently subjective value – it is almost impossible to estimate this via data or models. And so the quantiles we've used are subjective estimates for what we believe the current failings in JULES could be. The way we've set it up does indeed provide different values for the different grid cells, and the `mild bias' comes from the same Pearson and Tukey formula, which we've been more explicit about. (on what is now lines 271-283)

- Line 244: '100,000 different candidate parameter settings'. I think that these are only sampled over  $\theta$ ? And that the forcing variables are fixed for the selected s,t observation cases? – Is that correct?

Yes, have now clarified this is  $\vartheta$ . (on what is now line 284)

- Line 250: How are the 3000 non-implausible parameter settings to be run through JULES in the next wave selected from the 14475 that you have? Is there a way to ensure these are well-spaced over the non-implausible parameter space? Or, do you just target those with the highest emulator uncertainty? Or choose them randomly? This isn't clear.

We've added text here to clarify. To summarise, the 3000 subset settings are chosen using the same technique that was used to choose a subset of grid cells. This ensures some degree of good coverage in the non-implausible space. (on what is now lines 291-292).

- Line 250: '2000 chosen as in Section 3.1' I cannot see in Section 3.1 that co-ordinates are matched to parameter settings. Should this be 'as in Appendix C'?

It was meant to say Section 2.4, which we've fixed. We refer back to section 2.4, as that is "chronologically" when this was done first, but section 2.4 does itself then refer further down to the appendix for details.

- Line 266: '...very large values for GPP values, with the extremes >15gC/m2/day...' This should be '8-day averaged GPP', yes? Check here and throughout the the other sections that the output is described correctly. i.e. the raw daily GPP and the 8-day average GPP is not the same.

**Whilst it does say "/day", this is only the unit of the measurement. We have added a footnote making it clear that this is still 8-day average data.**

- Line 340 (with Line 233-235): '...explore many different values of the tuning parameters for a select few times and locations...'. How robust are the results in Section 3.3 to the selection of the observational data used for the comparison? Has this been tested?

We hadn't tested it before. For curiosity's sake, we have now re-run the process after getting this comment (note, this process is not fast). With the re-run (a new calibration grid cell selection and a new history match), we ruled-out 86.9% of the input space in a first wave. Interestingly, this is a more efficient rate of ruling-out parameter choices than the set of grid cells we had in the paper (85.5%), but it is not massively different.

The choice of observational data to try and match to obviously does affect the results to some degree. Part of this can also be due to emulator uncertainty – the emulator is not equally uncertain for all grid cells and so certain grid cells can be easier to match to, not because JULES is more accurate for these, but because emulator uncertainty is preventing ruling-out. In our case, wave 2 of the paper's history match leap-frogs the ruling-out percentage of this second choice of calibration grid cells – reaching 95.6% of the input space ruled out. It could be interesting to see if this second choice of grid cells would reclaim its lead in a second wave, but this would require additional supercomputer time, and is ultimately better left as future work.

A perfect solution might entail boosting the random number of coordinates used from 1000 to 10,000 (say), but this would have been significantly more computationally expensive (and very memory intensive).

- Line 372-373: 'The formulation only requires the inversion of m x m matrices...'. In the equation on line 372, the first matrix inversion is for ' $K_{nm}$ '. I think the subscript notation here needs to be amended to 'mm': so, ' $K_{mm}$ '. Also, I may be wrong, but this formulation looks to be following that in Equation 8 of Salimbeni and Deisenroth (2017) – please check the sign of the 'S' within the brackets in the equation on line 372, as in the Salimbeni paper Eq 8 this has opposite sign.

*Very good catch, thank you. For both points here. The equation has been amended. (on what is now line 440)*

**- Line 390: What is meant by 'mostly condensed'?**

We have removed the word "mostly". The reason it was included is because we do technically make use of the intra-grid cell variability in Equation B2 via the "C" term. However, this is a technicality, and as you point out it makes it less clear early on if we're ambiguous. Adding extra detail here (just a page before we clarify anyway) breaks the flow and isn't needed. As such, we've dropped down to just "condensed" (on what is now line 457).

- Line 393: 'The first score is the minimum distance between two  $w_s$  hat vectors'. Is this the minimum distance between any two  $w_s$  hat vectors from a selected potential set of these vectors? Please make this clearer in the text.

Yes, text has been amended to clarify. (on what is now line 461).

- Line 410-412: 'C is simply the difference...'. Is this calculated for each individual grid cell? If so, is it then summed over the grid cells in each potential set, and then summed again over the different forcing dimensions to produce a single number C for that set? Also, when summing over the forcing dimensions, are the values for each dimension scaled in some way so that a forcing quantity on a scale with larger magnitude isn't overweighted in the sum? Please clarify.

Yes it is summed over grid cell and dimension. And the forcing dimensions were all scaled to be between 0 and 1. Both are now clarified in the text (on what is now lines 479-480).

- Line 418-419: What is the dimension of  $\theta$  for these Latin hypercubes over  $\theta$ ? Please add this to the text. [I think it is 53 (all PFT parameters are treated separately here, and then the design will collapse down when fitting the emulators for each individual PFT, using just the PFT's subset of  $\theta$ ?)]

Yes it is 53. This is now mentioned (on what is now line 486)

**Technical Corrections: (These have all been done)**

- Line 60: Change 'to large' to 'too large'

- Line 79-80: The sentence: 'The simulator is treated as...' needs clarity, as at the start it says 'an uncertain function' for the simulator, but then this is referred to as 'those functions' on the next line in the same sentence.

- Line 135: Change '...in the appendix,... ' to '...in Appendix A,...'

- Line 159: 'See Appendix B for...' Should this be Appendix C?

- Tables at the top of Page 8: This is a formatting thing, but it looks weird for the end of Table 1 to come after Table 2 here? Please put Table 2 after Table 1.

- Line 174: Change 'combination' to 'combinations'

- Line 175: Change '...in the appendix' to '...in Appendix B,...' [Make it clear which section the reader should go to for the information.]

- Line 176: Remove the word 'also'.

- Line 178: '(see Appendix C for details).' Should this be Appendix D?

- Line 200: Replace ';' with ','

- Figure 2 caption, Line 3: Change: '...can be found in 1, and...' to '...can be found in **Table 1**, and...'

- Line 231: Change: '...,  $\theta$ , **that** can be ruled out.' to: '...,  $\theta$ , can be ruled out.'

- Line 242: There is a missing square bracket on the expectation of y term in the formula.

- Line 269: No need for a new paragraph here?

- Line 310: Should 'Appendix C' be changed to 'Appendix D'?

- Line 410: '...third score, C, is is simply...'. Remove the second 'is'.

- Remove the second 'appendix' from each of the Appendix sections titles. It isn't needed.

**Reviewer 2**

Thank you for you comments and suggestions. We believe the changes made have improved the article. What follows is a recap of the suggested changes, and how we have implemented them.

**Major Concerns:**

- (1) We get the full picture of the procedures involved only after reading the text. Since they are also presented in an extended manner, it is necessary to frequently go back to recall some of their components or figures. The inclusion of a flowchart describing the development of the emulator would be very helpful not only to give the whole picture to the readers since the beginning of the paper but also to show the location of each procedure in the whole sequence.

We have added a flow chart broadly outlining the different steps required to fit an emulator in our framework.

(2) I understand that the emulator can be used to extrapolate the model without extensive analyses, but I think it is necessary to discuss whether the emulator can be applied to other years instead of training data years. If an emulator is available, it would be interesting to discuss whether it is possible to replace the predictive simulations.

- The goal of an emulator is indeed to replace new simulations. For JULES and other land surface models, our methodology would allow for predictions for other years, other than just the training years. Our framework deals with each grid cell and timestep separately, and so the emulator is valid for any potential future grid cell and timestep that is not outside the ranges of our training years. In other words, the emulator is valid for any future year, so long as that year has properties that are not far beyond the extremes of our training years. How a land surface emulator performs with years that are particularly extreme would be an interesting direction for future research, and would likely warrant training the emulator using entirely, or partially, artificial driving data that covers any credible future land environment.

We have added a paragraph to the discussion section outlining this potential.

**(3) It is expected that calibration and tuning of the model will bring it closer to the observed data. However, I believe that it would be valuable for the paper to add a discussion on the advantages of the emulator over data assimilation methods.**

Data assimilation is a related technique, but it does serve a slightly different goal. Data assimilation aims to improve one specific simulation, making it closer to observed data. Tuning also aims to improve simulations, but it does so obtaining better values for the static input parameters. One can try and use data assimilation techniques to do tuning as well, but it is not clear if this is an effective strategy (Rougier 2013). We have added some discussion of data assimilation to Section 4.

**(4) I believe that the reader can better understand the usefulness of the emulator if Chapter 5 summarizes how this emulator can be applied.**

We've added a paragraph to section 4 explicitly making clear some of the applications of such an emulator. Emulators help with any goal where simulation runtime acts as a barrier to comprehensive analysis. Specifically, optimisation of land surface decisions is one interesting direction for future work.

**Minor Concerns**

**Line 177–178: Please add the year to Running and Zhao's citation.**

done

**In Fig. 3, the lines overlap and are not visible, and hence, the figure is illegible unless you devise a way to show the ensemble spread in a semi-transparent way by representing the ensemble averages of initial, windows1, and windows2 as solid lines.**

We have made the sampled ensemble members semi-transparent, which makes the individual lines more legible. We have also added text to the caption, making it explicitly clear that the red lines (if they were infinite) are a subset of the blue lines, which in turn (if they were infinite) are a subset of the grey lines). This information makes it more clear that the tuning procedure is shrinking the range of simulated GPP, and not simply moving it.

We have avoided adding ensemble averages or ensemble spread, as including these as well as the individual simulated ensemble members is not possible in a clear way, and we believe these summary statistics present an overly terse summary of the overall process to be presented alone.

**Line 311: I do not understand the intent in the following sentence: "Better observational data would improve the tuning; but the procedure itself does appear capable." Please explain this in more detail.**

**We have added to the text to clarify and expand.**

("Better observational data, with lower observational uncertainty and requiring fewer deletions / with less missing data, would likely further shrink the parameter space, improving the tuning results; but the overall tuning methodology itself appears capable.")

**Bibliography**

Rougier, Jonathan. 2013. "'Intractable and unsolved': some thoughts on statistical data assimilation with uncertain static parameters." *Philosophical Transactions of the Royal Society A: Mathematical, Physical and Engineering Sciences.*